# Energy Conversion and Entropy Production in Biased Random Walk Processes—From Discrete Modeling to the Continuous Limit

**DOI:** 10.3390/e25081218

**Published:** 2023-08-16

**Authors:** Henning Kirchberg, Abraham Nitzan

**Affiliations:** Department of Chemistry, University of Pennsylvania, Philadelphia, PA 19104, USA; anitzan@sas.upenn.edu

**Keywords:** thermodynamic process, entropy production, discrete state space, continuous state space, stochastic thermodynamics

## Abstract

We considered discrete and continuous representations of a thermodynamic process in which a random walker (e.g., a molecular motor on a molecular track) uses periodically pumped energy (work) to pass *N* sites and move energetically downhill while dissipating heat. Interestingly, we found that, starting from a discrete model, the limit in which the motion becomes continuous in space and time (N→∞) is not unique and depends on what physical observables are assumed to be unchanged in the process. In particular, one may (as usually done) choose to keep the speed and diffusion coefficient fixed during this limiting process, in which case, the entropy production is affected. In addition, we also studied processes in which the entropy production is kept constant as N→∞ at the cost of a modified speed or diffusion coefficient. Furthermore, we also combined this dynamics with work against an opposing force, which made it possible to study the effect of discretization of the process on the thermodynamic efficiency of transferring the power input to the power output. Interestingly, we found that the efficiency was increased in the limit of N→∞. Finally, we investigated the same process when transitions between sites can only happen at finite time intervals and studied the impact of this time discretization on the thermodynamic variables as the continuous limit is approached.

## 1. Introduction

Nonequilibrium thermodynamics deals with general laws of a (driven) system transferring energy from one or more heat bath(s) to useful work. The second law, however, restricts this transformation as only part of the input energy may be “accessible”, as the entropy production, related to heat production, must not decrease [1,2,3]. The “system” is usually described by many degrees of freedom, e.g., $10^{23}$ gas particles with individual fluctuating trajectories, where the exchanged heat and the extracted work are determined from statistical averages over all these particles trajectories [4]. For “small” systems, such as (bio)polymers, colloid particle, enzymes, or molecular motors, the dynamics is described by only few degrees of freedom and the fluctuation of individual “state” trajectories becomes more prominent [3,5,6]. Notably, it has been shown, e.g., see Refs. [7,8,9], that the resulting (relative) fluctuations of a statistical averaged observable, like the number of steps *R* of a molecular motor mimicked by a random walk along a track of molecules, can be related to the entropy production. A relative uncertainty or fluctuation $\Delta R^{2}=2D/v^{2}$ of the motor steps along the molecular track with the diffusion coefficient *D* and velocity *v* requires at least an entropy production rate $\dot{\sigma }$ of $2k_{B}/\Delta R^{2}$. This leads to the inequality, known as the thermodynamic uncertainty relation (TUR) [7,8]:(1)$$\begin{array}{c} \dot{\sigma }\frac{2D}{v^{2}}\geq 2k_{B}. \end{array}$$

Different approaches are used for describing small system dynamics including fluctuations and the resulting (stochastic) thermodynamic properties. Two of these approaches have been prominently elaborated in recent years [3]:

(i) Dynamics on a discrete set of states: A system is described by its microstates whose dynamics is captured by a master equation. The rates to interchange between microstates is governed by the local detailed balance relation determined by a thermal bath [3];

(ii) Dynamics on continuous trajectories: Consideration of individual continuous trajectories of a (driven) colloidal particle whose velocity is described by a Langevin equation where the probability distributions of a particle at position *x* with velocity *v* and diffusion *D* are determined by the Fokker–Planck equation.

Starting from discrete set of states, we can approximate the dynamics by a continuous description, such as the Fokker–Planck equation by standard procedures like the Kramers–Moyal expansion, when (infinite) many states may be visited during the time of observation, e.g., a chemical reaction network of multiple reaction steps [10,11,12,13,14]. Similarly, starting from a continuous description for the dynamics, a discrete representation is obtained by standard mathematical steps of replacing derivatives by finite difference quotients. While such mathematical transformations are expected to lead to equivalent descriptions of the underlying physics, we show below that not all physical observables can be kept invariant under such transformations. In particular, we show that the discrete to continuous limiting process is not unique and depends on which observables are chosen to be invariant under this process. Recent studies have already shown that the entropy production might differ under different coarse-graining schemes since, under coarse-graining, some “information” is lost while also the mathematical derivation of the differential entropy starting from the discrete Shannon entropy has revealed some discrepancy (Remark: Note that the Shannon entropy $S=-k_{B}\sum_{i}p_{i}\ln p_{i}$ is defined for a discrete propability distribution of *i* states (each state with the respective probability $p_{i}$). Going to the continuous state-space description for the probability distribution $p_{i}\to p\left(x_{i}\right)\Delta x$, the continuous entropy $s^{\Delta x}=-k_{B}\int dxp\left(x\right)\ln p\left(x\right)-lim_{\Delta x\to 0}k_{B}\sum_{i}p\left(x_{i}\right)\Delta x\ln\left(\Delta x\right)$ differs by a potentially infinite offset (since ln(Δx)→−∞ for Δx→0), which needs to be substracted, see discussion in [15]. However for the discussion on entropy production (change in entropy and its change per time (entropy production rate)), we note that the infinte offset vanishes) [15,16,17]. Here, we suggest a different point of view by showing that it is possible to impose an invariant entropy production (or a given heat exchanged with the thermal environment) when proceeding from the discrete to the continuous description of the system dynamics. Under such restriction, some other system observable cannot be kept invariant. Alternatively, one may even ask whether this observation may be translated into realistic system processes. Put differently, can we, by adding intermediate states between an initial and a target state, optimize the process of transferring input energy into useful work.

The aim of the present manuscript is to investigate the transition from a discrete to a continuous state-space system for an exactly solvable master equation by keeping distinct system observables constant while studying the impact on other observables. Explicitly, we consider a simple model: a Brownian particle moving on a downhill slope with an energy-pumping step taking place at constant length intervals *L* that restores the energy of the particle into its initial value, which represents a cycle. In the discrete representation, the particle is a random walker moving among *N* equally spaced sites per fixed length *L*. N→∞ represents the continuous limit for the cyclic dynamics. More details about the model are provided in Section 2. Other processes with period boundary conditions can be mapped into this form. One realization of such a process for walking in discrete steps is that of molecular motors, which transform free energy liberated in a chemical reaction by a succession of steps on a track into mechanical work (motion) [5,8]. Driven rotational Brownian particles through periodic potential wells to experimentally test thermodynamics laws represent another example, though there are many others [18]. Given the specific chosen conditions, like constant speed *v* and the entropy production rate of the cycle, we can study the diffusion coefficient *D* in dependence of the number of sites *N* per cycle and the continuous limit when N→∞. Interestingly, under an additional action of an opposing force during the process under study, we can also determine the impact of the number of sites per cycle on thermodynamic performance by transferring input energy into useful work. Furthermore, when keeping the energy drop per cycle and the speed constant, we interestingly found that, in the limiting case N→∞, the diffusion coefficient (equivalent to the variance in the site distribution) approaches the thermal Einstein relation when N→∞, which is usually only expected for small velocities in the linear response limit of small driving.

The paper is structured as follows. In Section 2, we describe the cyclic process as biased random between (energy) sites governed by a master equation restricted by periodic boundary conditions. We further discuss the implication for the entropy and heat production. In Section 3, we study the entropy production rate and heat transfer rate into the environment of the cyclic process given the velocity *v* and diffusion constant *D* for different number of sites *N* in the dynamical (relaxation from initial site) and in a steady state. We then determine the performance in transferring input power to useful output power for the cycle process under an opposing force by varying *N* and, so, by going from the discrete to the continuous (N→∞) state space. In Section 4, we study the transition to N→∞ given constant entropy production for the process, and, under either constant velocity *v* or diffusion coefficient *D*. The impact on the respective system variable when increasing *N* is discussed. Section 5 is devoted to examining the randomness parameter of the cycle, which is dependent on the number of sites *N*, while maintaining various physical observables as constants. Through this randomness factor, we can also determine the variance in the cycle completion time and study its dependence on *N*. In Section 6, we investigate the process by the same biased random walk, but where the system evolves at finite time intervals, and study the impact of this time discretization on the entropy production and diffusion coefficient. Section 7 concludes this work.

## 2. Cyclic Process

We consider a cyclic process, where the n=1,…,N sites are aligned on a circle such that each site has two neighboring sites with the periodic boundary condition N+n=n with N≥3. The process dynamics is captured by a biased one-step random walk with forward and backward transfer rates α and β, which are equal at each site, such that the classical (Markovian) master equation for the probability distribution of the *n*-site of the cycle reads [10,13,14]
(2)$$\begin{array}{c} \dot{P}\left(n,t\right)=\alpha P\left(n-1,t\right)+\beta P\left(n+1,t\right)-\left(\alpha +\beta \right)P\left(n,t\right). \end{array}$$
Starting from a well-defined site at t=0, one finds in the limit of long times t→∞ the (nonequilibrium) steady-state distribution for the *n* site to be P(n,t→∞)=1/N (see Appendix C). When writing the master Equation (Equation 2) in the form $\dot{P}=MP$, we see that the matrix *M* is irreducible with the dominant eigenvalue λ=0 while all other eigenvalues have a strictly negative real part according to the Perron–Frobenius theorem, signaling the existence of a stable steady-state vector [19,20]. While our cyclic process can take place in the state space of any given system, it is convenient to consider the equivalent random walk with forward and backward rates α and β on a cycle of constant circumference L=2πR. In this equivalent random walk problem, the velocity *v* and diffusion constant *D* at steady state are defined by [14]
(3)$$\begin{array}{c} v\equiv \frac{\langle n\rangle \Delta x}{t}=\left(\alpha -\beta \right)\Delta x \end{array}$$
and
(4)$$\begin{array}{c} 2D\equiv \frac{\left(\langle n^{2}\rangle -{\langle n\rangle }^{2}\right)\Delta x^{2}}{t}=\left(\alpha +\beta \right)\Delta x^{2}, \end{array}$$
where Δx=2πR/N is the equidistant step size, with *R* and *N* being the radius of the cycle and the number of total sites *N* of the cycle, respectively. Evidently, the time for completing a full cycle is $\tau =N{\left(\alpha -\beta \right)}^{-1}$.

The transition between the neighboring sites, n→n±1, are considered as autonomous Markov jump processes where each site has its energy E(n). Thermodynamic consistency is introduced by the local detailed balance condition
(5)$$\begin{array}{c} \frac{\alpha }{\beta }=e^{\frac{1}{k_{B}T}\left(E(n)-E(n+1)\right)}, \end{array}$$
where, for simplicity, we assume isothermal conditions with *T* on all sites, and where ΔE=E(n)−E(n+1)>0 is taken to be the same for all nearest neighbor sites. Note that we count the heat exchange with the bath as Q≡−ΔE, i.e., if Q<0, the amount of heat is taken from the system to the bath; while the environment provides heat to the system if Q>0. This implies that during a cycle, the amount of heat $\Delta Q\equiv -N\Delta E=k_{B}TN\ln\left(\frac{\alpha }{\beta }\right)$ is dissipated, so E(N+1)≡E(1)−NΔE may be compared to a downhill process of energy loss NΔE. To remain consistent with the periodic boundary conditions, we further assume that, between sites *N* and N+1=1, an upward energy jump occurs, in which the same amount of work, W=NΔE, is returned to the system.

Next, we calculate the time-dependent change of the Boltzmann–Gibbs entropy, $\dot{S}\left(t\right)=-k_{B}\sum_{n}\dot{P}\left(n,t\right)\ln P\left(n,t\right)$, [2,21] for the system using the master Equation (Equation 2). We obtain
(6)$$\begin{array}{cc} \dot{S}\left(t\right) & =-k_{B}\sum_{m=1}^{N}\sum_{n=1}^{N}[\alpha P\left(n-1,t\right)+\beta P\left(n+1,t\right)\\ & -\left(\alpha +\beta \right)P\left(n,t\right)]\ln\frac{P(n,t)}{P(m,t)} \end{array}$$
(7)$$\begin{array}{c} =k_{B}\sum_{m=1}^{N}\sum_{n=m-1}^{N-1}\left[\alpha P(n,t)-\beta P(m,t)\right]\ln\frac{P(n,t)}{P(m,t)} \end{array}$$
(8)$$\begin{array}{cc} & =k_{B}\sum_{m,n=m-1}\left[\alpha P(n,t)-\beta P(m,t)\right]\ln\frac{P(n,t)\alpha }{P(m,t)\beta }\\ & -k_{B}\sum_{m,n=m-1}\left[\alpha P(n,t)-\beta P(m,t)\right]\ln\frac{\alpha }{\beta }. \end{array}$$
Note that in Equation (Equation 6), we could replace lnP(n,t) by ln[P(n,t)/P(m,t)] because the sum that multiplies lnP(m,t) vanishes. The result (Equation 8) can be recast in the form $\dot{S}\left(t\right)={\dot{S}}_{e}+\dot{\sigma }\left(t\right)$, [22,23] where, using $\sum_{n}P\left(n,t\right)=1$,
(9)$$\begin{array}{cc} {\dot{S}}_{e} & =-k_{B}\sum_{m,n=m-1}\left[\alpha P(n,t)-\beta P(m,t)\right]\ln\frac{\alpha }{\beta }\\ & =-k_{B}\left[\alpha -\beta \right]\ln\frac{\alpha }{\beta }, \end{array}$$
and
(10)$$\begin{array}{cc} \dot{\sigma }\left(t\right) & =k_{B}[\left[\alpha -\beta \right]\ln\frac{\alpha }{\beta }\\ & +\sum_{m,n=m-1}\left[\alpha P(n,t)-\beta P(m,t)\right]\ln\frac{P(n,t)}{P(m,t)}]. \end{array}$$

Next, we show that the first term ${\dot{S}}_{e}$ (Equation (Equation 9)) is the entropy flow into the environment, while the second term is the entropy production rate $\dot{\sigma }\left(t\right)$ (Equation (Equation 10)).

Consider first Equation (Equation 9). Because the rates α and β were assumed not to depend on the site identity, ${\dot{S}}_{e}$ is time-independent. ${\dot{S}}_{e}$ can be written as
(11)$$\begin{array}{c} {\dot{S}}_{e}=J\frac{Q}{T}, \end{array}$$
where J≡α−β is the cumulative flux: the sum over nearest neighbor site-pair fluxes (see first expression in Equation (Equation 9)). *Q* is the heat exchanged during a single nearest neighbor transfer event with the environment according to the local detailed balance relation in Equation (Equation 5). The product JQ in Equation (Equation 11) is the heat flux into the thermal environment of temperature *T* per cycle. For the given model, this heat exchange is time-independent.

According to the Clausius principle, the change in system entropy is bounded by the (negative) heat amount exchanged with the environment $\dot{S}\geq \frac{JQ}{T}$ where equality is reached for reversible processes [1]. Motivated by this inequality, one defines the total entropy production rate by $\dot{\sigma }\left(t\right)=\dot{S}-\frac{JQ}{T}\geq 0$. Indeed, the second term $\dot{\sigma }\left(t\right)$ (Equation (Equation 10)) meets the two important properties: (i) It is non-negative because the first term in Equation (Equation 8) can be recast into (x−y)ln(x/y)≥0; and (ii) it vanishes for thermal equilibrium, when microscopic reversibility or the detailed balance condition, αP(m,t)=βP(n,t), is obeyed and no entropy is produced. In a (nonequilibrium) steady state, Equation (Equation 6) is zero and the entropy production equals the negative entropy flow into the environment $\dot{\sigma }=-{\dot{S}}_{e}$, Refs. [23,24].

In the following, we investigate the entropy production rate $\dot{\sigma }\left(t\right)$ and the physical measurable entropy or heat flow ${\dot{S}}_{e}$ into the environment for the prototype *N*-site cyclic process given different conditions. In particular, we are interested in the limit N→∞ to investigate the transition from the discrete to the continuous state space.

## 3. N-Site Cyclic Process under Constant Velocity *v* and Diffusion Constant *D*

We first consider the dependence of the number of sites *N* per cycle on the system that is performed under the conditions that (a) the speed *v* (Equation (Equation 3)) and (b) the diffusion constant *D* (Equation (Equation 4)) are kept constant. The first condition requires that the time, $\tau =N{\left(\alpha -\beta \right)}^{-1}$, for a full completion of the cycle remains constant. Together with the second condition, the forward and backward rate must depend on *N*. We obtain
(12)$$\begin{array}{cc} \alpha & =\frac{DN^{2}}{4\pi^{2}R^{2}}+\frac{vN}{4\pi R}, \end{array}$$
(13)$$\begin{array}{cc} \beta & =\frac{DN^{2}}{4\pi^{2}R^{2}}-\frac{vN}{4\pi R}. \end{array}$$
To ensure the positivity of the rate (13), β≥0, it immediately follows that v≤ND/Rπ. Note that under the conditions restricted by Equations (Equation 12) and (13), the detailed balance relation $\Delta E=k_{B}T\ln\left(\frac{\alpha }{\beta }\right)$ (Equation (Equation 5)) will be a nonlinear function of *N*.

Consider the entropy production rate $\dot{\sigma }\left(t\right)$ (Equation (Equation 10)) given a well-defined initial cycle site n=1 at t=0, so P(n=1,t=0)=1. The entropy production rate (the entropy produced per unit time) decreases over time before reaching a (*N*-dependent) steady-state value, see Figure 1.

This observation can be understood as follows. Initially, the system starts at a given site and evolves in its cyclic dynamics according to the master Equation (Equation 2). At later time *t*, the system will be found at a given site *n* with probability P(n,t) (see Appendix A, Appendix B and Appendix C). The loss of the initial “knowledge” about the exact system site *n* increases the entropy of the system. As the probability distribution relaxes to its steady-state distribution $P_{n,SS}\equiv P\left(n,t\to \infty \right)=1/N$ (see Appendix C), the entropy production rate, $\dot{\sigma }$, decreases to its steady-state value
(14)$$\begin{array}{cc} {\dot{\sigma }}_{SS} & =k_{B}\left(\alpha -\beta \right)\ln\left[\frac{\alpha }{\beta }\right]=\frac{k_{B}N}{2\pi R}v\ln\left[\frac{\frac{2DN}{2\pi R}+v}{\frac{2DN}{2\pi R}-v}\right]. \end{array}$$

Note that entropy is produced at constant rate when running the cyclic process under steady-state conditions. Interestingly, as seen in Figure 1, the more sites *N* are included in the cycle of finite length the faster the entropy production rate decreases to its steady-state value. At steady state, the entropy production equals the negative (in this model time-independent) entropy flow into the environment ${\dot{\sigma }}_{SS}=-{\dot{S}}_{e}$.

Consider the *N*-dependence of the the steady-state entropy production rate ${\dot{\sigma }}_{SS}$ for an *N*-site cycle, Equation (Equation 14). The dependence is shown in Figure 2. For N→∞, the steady-state entropy production rate (Equation 14) is reduced to $k_{B}v^{2}/D$. This minimum entropy production rate in this limit can be understood as the forward and backward rates (Equations (Equation 12) and (13)) become more alike. Therefore, consecutive transitions between sites *n* and n+1 become more time-symmetric and less entropy per step, ΔE/T=ln[α(N)/β(N)]/T, is produced.

Interestingly, for N→∞, the TUR relation, the minimal required entropy production rate $\dot{\sigma }$ for a given relative fluctuation $2D/v^{2}$ (as introduced in Equation (Equation 1) for a molecular motor moving along a molecular track) states equality, $\dot{\sigma }2D/v^{2}=2k_{B}$. In this limit, the dynamics of the cyclic process is comparable to a continuous Brownian diffusive motion of a particle of constant speed *v* and diffusion *D* described by a Fokker–Planck equation [25].

Assume that a forward (downhill) step on the cycle takes place against a constant applied force *f*. Then, the local detail balance relation (Equation 5) must be redefined by $\Delta \tilde{E}\equiv E\left(n\right)-E\left(n+1\right)-f\Delta x=\Delta E-f\Delta x>0$, as part of the energy per step ΔE is transferred to work fΔx, where Δx is the step size. We defined the system’s heat exchange per step with the bath as $Q\equiv -\Delta \tilde{E}$, while Q<0 is the amount of heat taken from the system to the bath. At steady state, the heat flow from the system into the bath is $-\dot{Q}=T{\dot{\sigma }}_{SS}$, see discussion in Section 2. The power output per cycle when running against the force at steady state is
(15)$$\begin{array}{cc} P\equiv \sum_{m,n=m-1} & \left[\alpha P(n)-\beta P(m)\right]f\Delta x=\left(\alpha -\beta \right)\Delta xf\\ & =vf. \end{array}$$
In Equation (Equation 15), αP(n)−βP(n+1) is the probability flux between neighboring sites and where $\sum_{n}P\left(n\right)=1$. By the first law of thermodynamics, the total supplied power must be $P-\dot{Q}$ such that we can define the thermodynamic efficiency as [8]
(16)$$\begin{array}{cc} \eta (N) & \equiv \frac{P}{P-\dot{Q}}=\frac{fv}{fv+T{\dot{\sigma }}_{SS}\left(N\right)}, \end{array}$$
where ${\dot{\sigma }}_{SS}\left(N\right)$, Equation (Equation 14), depends on the total number of sites *N* per cycle. The cycle can be compared to a process of going down a slope against a constant force and with friction. The friction force is usually taken as $F_{fr}=\gamma v$ such that the related heat dissipated in the cycle per unit time is $F_{fr}v=\gamma v^{2}$. By identifying $\gamma v^{2}\equiv T{\dot{\sigma }}_{SS}\left(v,N\right)$, we can calculate the friction coefficient $\gamma \left(v\right)=T{\dot{\sigma }}_{SS}\left(v\right)/v^{2}$ for the present cycle process using Equation (Equation 14). Note that, as expected, the friction coefficient γ goes to its linear response value for a small velocity, $\gamma \left(v\to 0\right)\to k_{B}T/D$, independent of *N*. Surprisingly, as depicted in the inset of Figure 3, γ also takes the same value for finite *v* in the limit of N→∞. In this limit, the system assumes some features similar to equilibrium even though the flux is finite. Not only in the limit v→0, but also in the limiting process can N→∞ be compared to thermodynamic cycles where the system changes adiabatically slowly to always be in thermal equilibrium throughout the process.

Equivalently, in the limit N→∞, we can write the thermodynamic efficiency (Equation 16) as
(17)$$\begin{array}{c} \eta =\frac{1}{1+k_{B}Tv/Df}. \end{array}$$

Note that in the limit of linear response the velocity *v* and force *f* are linearly related by the mobility μf=v. Assuming the Einstein relation $D=\mu k_{b}T$ holds under linear response such that the efficiency in the limit N→∞ reach η=1/2.

Figure 3 portrays the efficiencies for different *N*. As expected, the efficiency increases with *N* as less heat $-\dot{Q}=T{\dot{\sigma }}_{SS}$ will be produced per cycle given constant power output *P*. Interestingly, the slower the chosen velocity *v*, the more one can reach maximal efficiency. The efficiency is bound from above, η≤1, while equality is reached for v=0 or f→∞, D→∞, see Equation (Equation 17). The last conditions, however, do not produce useful output power, as the cycle will stop.

## 4. Cyclic Process with Constant Velocity *v* or Diffusion Constant *D* and Constant Energy Drop Per Cycle

As stated in the introduction, the process of going to the continuous description for the state-space dynamics (N→∞) is not unique. We, therefore, use the same methodology as above to describe the unicyclic process as a 1D random walk at steady state but where we now require the total entropy produced per cycle to be constant (given by the constant energy drop per cycle) and either (A) the velocity *v* or (B) the diffusion coefficient *D* to be constant. We assume that all energy invested into the system is dissipated as heat to the environment, so W=σT. The steady-state entropy production σ is
(18)$$\begin{array}{c} \sigma =k_{B}N\ln\left(\frac{\alpha }{\beta }\right), \end{array}$$
where *N* is the total number of sites. At steady state, the entropy production equals the heat going into the environment, see Section 2.

(A) Given the constant velocity *v*, Equation (Equation 3), the forward and backward rates are
(19)$$\begin{array}{cc} \alpha & =\frac{vN}{2\pi R\left(1-e^{\frac{-\sigma }{Nk_{B}}}\right)}, \end{array}$$
(20)$$\begin{array}{cc} \beta & =\frac{vN}{2\pi R\left(e^{\frac{\sigma }{Nk_{B}}}-1\right)}, \end{array}$$
where *R* is the radius of the cycle (see Section 2).

(B) Given the constant diffusion coefficient *D*, Equation (Equation 4), the forward and backward rates are
(21)$$\begin{array}{cc} \alpha & =\frac{2DN^{2}}{{\left(2\pi R\right)}^{2}\left(1+e^{\frac{-\sigma }{Nk_{B}}}\right)}, \end{array}$$
(22)$$\begin{array}{cc} \beta & =\frac{2DN^{2}}{{\left(2\pi R\right)}^{2}\left(e^{\frac{\sigma }{Nk_{B}}}+1\right)}. \end{array}$$

Consider first case (A). Equation (Equation 4) with Equations (Equation 19) and (20) leads to
(23)$$\begin{array}{cc} D & =\frac{1}{2}\left(\alpha +\beta \right){\left(\frac{2\pi R}{N}\right)}^{2}=\frac{v\pi R}{N}\coth\left[\frac{\sigma }{2Nk_{B}}\right], \end{array}$$
which is shown in Figure 4. The diffusion coefficient decreases with increasing *N*. As the diffusion coefficient is the variance of the site distribution on our equivalent cycle, see Equation (Equation 4), the related fluctuations in localization of a site is reduced during a cycle with an increase in *N*. Assuming that the energy falls uniformly along the cycle, so that ΔE=Tσ/N, we find that, in the regime of linear response v→0 and in the limit N→∞, the diffusion coefficient is captured by the (Einstein) relation $D=v/Fk_{B}T=\mu k_{B}T$ with the mobility μ=v/F and the related force F=Tσ/2πR in analogue to the friction coefficient (as discussed in Section 3) [26,27].

Next, consider case (B). Equation (Equation 3) with Equations (Equation 21) and (22) leads to the velocity on *N*
(24)$$\begin{array}{c} v=\left(\alpha -\beta \right)\frac{2\pi R}{N}=\frac{DN}{\pi R}\tanh\left[\frac{\sigma }{2Nk_{B}}\right]. \end{array}$$
The velocity *v* increases with *N* and so the time of a full completion of the system cycle τ=2πR/v is reduced and minimizes for N→∞, see Figure 5.

## 5. Randomness Parameter and Variance in Cycle Completion Time

In the previous Section 3 and Section 4, we analyzed, for a biased random process, the dependence of different physical observables on the number intermediate sites *N* taken to complete a given cycle of operation, as a way to demonstrate the nonuniqueness of going to the continuous limit (N→∞) of this process. Another perspective of this problem is studied in Refs. [28,29,30,31], where the values of physical observables associated with enzyme-catalyzed cycles were used to set bounds on the number of intermediate cycle steps (Note that in difference to enzymic cycles where the step size is considered as constant, we reduce the stepsize Δx=L/N in a cycle of finite lenght L=2πR with increasing *N* to study the impact of the limit N→∞ on the physical observables). It was pointed out that, in addition to the average speed *v* and diffusion coefficient *D*, their ratio provides important information on the observed walk statistics [28,29,30,31]. Explicitly, we characterize the random process in terms of their forward and backward rates, α and β, respectively, and an equidistant step length Δx. If the process starts at n=1 and a random site *n* is reached at time *t*, then the process will be on average at site 〈n〉=vt/Δx, whilst the random diffusive process produces a variance in the site by $\langle \delta n^{2}\rangle \equiv \langle n^{2}\rangle -{\langle n\rangle }^{2}=2Dt/\Delta x^{2}$, see Equations (Equation 3) and (Equation 4) in the limit t→∞. These two quantities can be combined into a randomness parameter, which, for the given step size Δx, reads [28,29,30,31]
(25)$$\begin{array}{c} r\equiv \frac{\langle \delta n^{2}\rangle }{\langle n\rangle }=\frac{2D}{\Delta xv}=\frac{\alpha +\beta }{\alpha -\beta }, \end{array}$$
where *v* and *D* are defined by Equations (Equation 3) and (Equation 4), respectively. Alternatively, we may consider the random passage time τ at which, starting from n=1, the walk reached the site *N* for the first time, namely, a distance NΔx from the starting point. For walks of uniform step length and finite bias, it has been shown [28,29] that, for a large enough *N*, the randomness parameter can be expressed in terms of the first two moments of the passage time distribution
(26)$$\begin{array}{c} r=\frac{\langle \tau^{2}\rangle -{\langle \tau \rangle }^{2}}{{\langle \tau \rangle }^{2}}=\frac{\langle \delta \tau^{2}\rangle }{{\langle \tau \rangle }^{2}}, \end{array}$$
where 〈τ〉 is the average time for a cycle completion and $\langle \delta \tau^{2}\rangle$ is its variance. Note that for many enzyme reaction cycles, the backward reaction rates are often sufficiently low as to be negligible. In such cases, the pathway consists of a sequence of *N* forward reactions only and the randomness parameter ($r=N_{min}^{-1}$) can be used to estimate the minimal number of kinetic sites that compose the underlying kinetic model [29,31]. In general, when considering forward and backward steps and using the average cycle completion time 〈τ〉, we can calculate the variance in cycle completion time to
(27)$$\begin{array}{c} \langle \delta \tau^{2}\rangle =r{\langle \tau \rangle }^{2}. \end{array}$$

We can now apply the results of Section 3 and Section 4 to examine the behavior of these observables in our different limiting cases. In our case, *N* corresponds to the number of sites per cycle and, consequently, τ is the time for the process to complete the cycle. Increasing *N* is achieved by eventually approaching a continuous description, so the cycle length NΔx=2πR is kept fixed.

Consider first the condition of a constant velocity and diffusion constant by increasing the number of sites *N*, see Section 3. Using Equations (Equation 12) and (13) in (Equation 25), the randomness parameter is
(28)$$\begin{array}{c} r=\frac{D}{\pi Rv}N \end{array}$$
and is linear in *N*.

Next, for the condition of constant entropy production σ per cycle under either constant velocity *v* or constant diffusion *D*, see Section 4, we find in both cases, using the respective rates in Equations (Equation 19) and (20) or Equations (Equation 21) and (22), the randomness parameter to be
(29)$$\begin{array}{c} r=\coth\left[\frac{\sigma }{2Nk_{B}}\right]. \end{array}$$
The randomness parameter (Equation 29) is determined by the number of sites *N* and the thermodynamic entropy production or heat dissipation into the environment, which equals, when neglecting the movement against an external force, the energy drop per cycle, see Section 2. Note that *r* tends to infinity in the limit σ→0 and in the continuous limit N→∞ since both limits reflect the equilibrium situation where forward and backward rates will be alike. In the limit σ→∞, given finite *N*, we find r=1, which is expected for the so-called “Poisson” motion since the infinite energy drop per cycle leads to an unidirectional motion [28].

With the randomness parameter at hand, we can now study the variance in the cycle completion time (Equation (Equation 27)). It has been shown that, for a biased random walk for a large *N*, the average completion time $\langle \tau \rangle =N{\left(\alpha -\beta \right)}^{-1}$ [32]. Consider first the cases (A) of keeping the velocity and diffusion coefficient constant and (B) keeping the entropy production per cycle and velocity constant. We find for (A) the given Equation (Equation 28), together with Equations (Equation 12) and (13) in Equation (Equation 27), where $\langle \tau \rangle =N{\left(\alpha -\beta \right)}^{-1}$, the variance in cycle completion time to be
(30)$$\begin{array}{c} \langle \delta \tau^{2}\rangle =\frac{4D\pi R}{v^{3}}N, \end{array}$$
and, equivalent for case (B), by using Equation (Equation 29), together with Equations (Equation 19) and (20) in Equation (Equation 27)
(31)$$\begin{array}{c} \langle \delta \tau^{2}\rangle =\coth\left[\frac{\sigma }{2Nk_{B}}\right]\frac{4\pi^{2}R^{2}}{v^{2}}. \end{array}$$
In both Equations (Equation 30) and (Equation 31), the variance in the cycle completion time increases monotonously with the site number *N*. This reflects the fact that, with an increasing number of sites *N* per cycle, the intersite rates become more alike, which increases the overall “randomness”, and, thus, $\langle \delta \tau^{2}\rangle$ for the total cycle completion.

In contrast, when *N* is changed while keeping a constant diffusion coefficient and entropy production, Equation (Equation 26), together with (Equation 24) and (Equation 29) lead to
(32)$$\begin{array}{c} \langle \delta \tau^{2}\rangle =\coth^{3}\left[\frac{\sigma }{2Nk_{B}}\right]\frac{4{\left(\pi R\right)}^{4}}{D^{2}N^{2}}. \end{array}$$
Interestingly, $\langle \delta \tau^{2}\rangle$ goes through a minimum with increasing site number *N*, see Figure 6. It should be kept in mind, however, that Equation (Equation 26) and, consequently, Equation (Equation 27), were derived under the assumption that *N* is large so that this observation should not be regarded as conclusive.

## 6. N-Site Process with Time Step Discretization

Consider now the same biased random walk process in which a cycle of length 2πR is traversed in *N* steps, so that NΔx=2πR, but where the system is restricted to move (by intersite distance Δx) only at finite time intervals Δt. Indeed, small systems which are periodically driven can be thought of as discrete-time processes, see [33]. As shown below, Δx and Δt are not independent of each other but some freedom exists in their choices. NΔE is the energy drop per such cycle (see Section 2, and recall that ΔE determines the detailed balance ratio of the forward and backward rates according to Equation (Equation 5)). The probability to be at site *n*, namely, at position x=nΔx on the cycle at time t=MΔt, is governed by the Makrov chain (Note that in contrast to other descriptions of random walks in forms of a discrete Markov chain, e.g,. for waiting time distributions [34], where after a time interval, sampled from such a distribution, a “jump” always happens with constant probabilities *p* and 1−p for a forward backward jump, respectively, here, the “walker” can also remain at its original position after Δt and where the probabilities depend linearly on Δt) [34]
(33)$$\begin{array}{cc} P(x,t+\Delta t) & =\alpha \Delta tP(x-\Delta x,t)+\beta \Delta tP(x+\Delta x,t)\\ & +(1-\alpha \Delta t-\beta \Delta t)P(x,t). \end{array}$$

Here, αΔt and βΔt are the probabilities (both assumed linear in Δt) to move a step forward and backward, respectively. Note that (α+β)Δt≤1 has to be imposed in Equation (Equation 33) to ensure positivity. As before, the process has periodic boundaries so that, after the final site n=N has been reached, it restarts at the beginning n=1 and its original energy is restored by some external work reservoir between sites *N* and N+1=1 (see discussion in Section 2). We use this model to study the effect of time discretization on the dynamical properties of the process. To calculate the velocity *v* and diffusion coefficient *D*, we determine the generating function $P\left(s,t\right)=\sum_{x}s^{x}P\left(x,t\right)$ where the moments can be calculated by $\langle x^{m}\rangle ={\left(s\partial /\partial s\right)}^{m}P\left(s,t\right){|}_{s=1}$. For the initial condition P(x=0,t=0)=1 (so that P(s,t=0)=1), we find the generating function to be
(34)$$\begin{array}{cc} P(s,t= & M\Delta t)\\ & ={\left[\alpha \Delta ts^{\Delta x}+\beta \Delta ts^{-\Delta x}+\left(1-\alpha \Delta t-\beta \Delta t\right)\right]}^{M}. \end{array}$$

The velocity and diffusion coefficient are determined as follows. Starting at x=0, we find in the long time limit (at steady state) t→∞ (see details in Appendix D)
(35)$$\begin{array}{cc} v & =\frac{\langle x\rangle }{t}=\left(\alpha -\beta \right)\Delta x; \end{array}$$
(36)$$\begin{array}{cc} 2D & =\frac{\langle x^{2}\rangle -{\langle x\rangle }^{2}}{t}=\left(\alpha +\beta \right)\Delta x^{2}-{\left(\alpha \Delta x-\beta \Delta x\right)}^{2}\Delta t\\ & =\left(\alpha +\beta \right)\Delta x^{2}-v^{2}\Delta t. \end{array}$$

The velocity *v*, Equation (Equation 35), is the same as in the continuous time case (Equation (Equation 3)), whereas the diffusion coefficient in Equation (36) is smaller by $v^{2}\Delta t$ in comparison to the continuous time case of Equation (Equation 4). Refs. [35,36] associate the bigger variance in the continuous-time master equation with higher fluctuations in the total number of hops observed in a given time interval. Next, consider the process as *N* increases. As in Section 3 and Section 4, we may consider an increase in *N* while keeping *v* and *D* constant or while keeping only one of them together with NΔE constant. As examples of the effect of moving in discrete time steps, we study the cases (A) constant *v* and *D* and (B) constant *v* and NΔE.

(A) Keeping *v* and *D* constant, we scale the rates again analog to Equations (Equation 19) and (20) with the total site number *N* (given by the intersite distance Δx=2πR/N)
(37)$$\begin{array}{cc} \alpha & =\frac{\left(D+v^{2}\Delta t/2\right)N^{2}}{{\left(2\pi R\right)}^{2}}+\frac{vN}{4\pi R}, \end{array}$$
(38)$$\begin{array}{cc} \beta & =\frac{\left(D+v^{2}\Delta t/2\right)N^{2}}{{\left(2\pi R\right)}^{2}}-\frac{vN}{4\pi R}. \end{array}$$
Note that the modification of the rates α and β depends on Δt. As before, this rescaling also implies a change in ΔE (see discussion in Section 3) so that the detailed balance relation is maintained. The condition (α+β)Δt≤1 in Equation (Equation 33) together with Equations (Equation 37) and (38) restricts the choices for Δt given Δx=2πR/N to
(39)$$\begin{array}{c} 0\leq \Delta t\leq \frac{-D}{v^{2}}+\sqrt{\frac{D^{2}}{v^{4}}+\frac{\Delta x^{2}}{v^{2}}}, \end{array}$$
which implies that Δt and Δx cannot be assigned independently of each other. In the limit N→∞ (Δx→0), this inequality (Equation (Equation 39)) becomes $0\leq \Delta t\leq \Delta x^{2}/2D$.

Next, consider the entropy production for this discrete hopping process. The average entropy change per step is
(40)$$\begin{array}{c} \Delta \sigma =k_{B}\alpha \ln\left[\frac{\alpha }{\beta }\right]+k_{B}\beta \ln\left[\frac{\beta }{\alpha }\right], \end{array}$$
where the two terms represent the entropy change in a forward and backward step multiplied by the probabilities that the respective step occurs. The rate of entropy change at steady state (the entropy production rate) is given by
(41)$$\begin{array}{cc} {\dot{\sigma }}_{SS} & =k_{B}\frac{\Delta \sigma }{\Delta t}=k_{B}\left(\alpha -\beta \right)\ln\left[\frac{\alpha }{\beta }\right]\\ & =\frac{k_{B}N}{2\pi R}v\ln\left[\frac{\frac{\left(D+v^{2}\Delta t/2\right)N^{2}}{{\left(2\pi R\right)}^{2}}+v\frac{N}{4\pi R}}{\frac{\left(D+v^{2}\Delta t/2\right)N^{2}}{{\left(2\pi R\right)}^{2}}-v\frac{N}{4\pi R}}\right]. \end{array}$$
Interestingly, comparing the resulting expression (Equation 41) to its analog (Equation 14) for the continuous master equation, we obtain a similar result, but with an additional term $v^{2}\Delta t/2$ added to the diffusion constant. The additional term effectively modifies the TUR relation (Equation (Equation 1)) as the relative uncertainty $2D/v^{2}$ changes. A similar observation was made in Ref. [37]. The resulting entropy production rate in Equation (Equation 41) for a given site number *N* per cycle is reduced if we choose a finite Δt (given the restriction on choices of Δt by Equation (Equation 39)), see Figure 7. This might be understood since, during a given time interval, the variance in position *x* on the cycle is reduced ((Equation (36)) by allowing intersite hops only in intervals Δt. In the continuous limit (N→∞, Δx→0 and Δt→0), however, Equation (Equation 41) yields
(42)$$\begin{array}{c} {\dot{\sigma }}_{SS}=k_{B}\left[\frac{v^{2}}{D}\right], \end{array}$$
which is the same result as that obtained in this limit in Section 3 (see Equation (Equation 14)).

(B) Next, consider the dependence on *N* under the condition of a constant energy drop NΔE per cycle and constant velocity *v*. This is the analogue consideration as in Section 4, but where the intersite hops are only allowed at time intervals Δt (which are restricted by the given *N* according to Equation (Equation 46) below). We assume that all energy invested into the system is dissipated as heat into the environment, so the steady-state entropy production per cycle is
(43)$$\begin{array}{c} \sigma =N\frac{\Delta E}{T}=k_{B}N\ln\left(\frac{\alpha }{\beta }\right), \end{array}$$
where *N* is the total number per cycle.

Given the constant velocity, Equation (Equation 35), the forward and backward rates are equivalent to Equations (Equation 19) and (20): (44)$$\begin{array}{cc} \alpha & =\frac{vN}{2\pi R\left(1-e^{\frac{-\sigma }{Nk_{B}}}\right)}, \end{array}$$(45)$$\begin{array}{cc} \beta & =\frac{vN}{2\pi R\left(e^{\frac{\sigma }{Nk_{B}}}-1\right)}. \end{array}$$
The restriction (α+β)Δt≤1 in Equation (Equation 33), together with Equations (Equation 44) and (45), limits the choices for Δt given Δx=2πR/N to
(46)$$\begin{array}{c} 0\leq \Delta t\leq \frac{\Delta x}{v}\tanh\left[\frac{\sigma }{2Nk_{B}}\right]. \end{array}$$
In the limit N→∞ (Δx→0 and Δt→0), the inequality in Equation (Equation 46) becomes $0\leq \Delta t\leq \Delta x\sigma /(vNk_{B})$.

For a given *N* and Δx=2πR/N, the time step Δt needs to satisfy the inequality (Equation 46). Here, we take
(47)$$\begin{array}{c} \Delta t=a\frac{\Delta x}{v}\tanh\left[\frac{\sigma }{2Nk_{B}}\right] \end{array}$$
with 0≤a≤1 and use Equation (36) to obtain
(48)$$\begin{array}{cc} D & =\frac{1}{2}\left(\left(\alpha +\beta \right)\Delta x^{2}-v^{2}\Delta t\right)\\ & =\frac{v\pi R}{N}\left(\coth\left[\frac{\sigma }{2Nk_{B}}\right]-a\tanh\left[\frac{\sigma }{2Nk_{B}}\right]\right). \end{array}$$
Interestingly, the dependence on time discretization translates here to a dependence of *D* on the choice of *a*, see Figure 8. Given *N* for different finite time Δt (scaled between 0≤a≤1), the diffusion coefficient, and consequently, the variance in x=nΔx, are strongly reduced for increasing Δt (less fluctuations in the total number of transitions for finite Δt). As expected, in the limit N→∞ and Δt→0, the diffusion constant takes the form $D=v/Fk_{B}T=\mu k_{B}T$ with the mobility μ=v/F and the corresponding force F=Tσ/2πR, [26,27], see discussion in Section 4. Therefore, given the chosen Δt and its above-discussed effect on *D*, the latter increases or decreases with *N* to the final value $D=\mu k_{B}T$ as depicted in Figure 8.

To summarize this section, when describing the dynamics of a (cyclic) process in discrete time intervals, the thermodynamic properties of the process, e.g., the entropy production or diffusion coefficient, are affected by this time discretization. The discretization in time, however, cannot be chosen arbitrarily but must obey the bounds given by the state-space discretization of the process. If the process dynamics become continuous state-space dynamics (N→∞), the time evolution needs to be described by intervals Δt→0, i.e., equivalently by a continuous time scale and all effects vanish.

## 7. Conclusions

In this paper, we investigate a cyclic and unithermal thermodynamic process using a model of a biased random walk between *N* sites on a cycle of a given length with an exact solvable master equation. We note that many dynamical site (state)-space processes with periodic boundary conditions may be equivalently mapped to this process. The limit N→∞ corresponds to the continuous limit that is usually captured by a Fokker–Planck equation. This limit is taken by keeping low order moments of system observables, i.e., the velocity *v* and diffusion coefficient *D*, constant. We show that the entropy produced, or, equivalently, the energy drop per cycle, is reduced when moving towards this continuous description. This has direct consequences for the efficiency of transferring input power into useful output power when an opposing force acts on the cyclic process. In particular, more power can be extracted from the process with an increasing number of sites *N* per cycle length.

An important outcome of our analysis is that the procedure in going to the continuous description of the process is not unique and depends on the physical observables that are assumed to be invariant under this limiting process. In addition to taking the limit N→∞ while keeping *v* and *D* constant, we also analyzed this limiting process while keeping *v* or *D* and NΔE constant. Interestingly, considering the limiting process under constant *v* and NΔE, we show that the diffusion coefficient *D* for a finite cycle velocity *v* in the limit N→∞ takes the same value as in linear response v→0 limit. Additionally, when analyzing the cycle randomness statistics and, in particular, the variance in the cycle completion time $\langle \delta \tau^{2}\rangle$, we found that, with increasing *N*, $\langle \delta \tau^{2}\rangle$ increases, signaling the increasing randomness in the cycle. Finally, we studied the dependence on *N* in the case where the transitions between sites are only allowed at fixed time intervals Δt. We found that not only the entropy production rate per cycle (when *v* and *D* are kept constant), but also the diffusion coefficient *D* (when *v* and NΔE are kept constant), are strongly affected by the way time discretization is introduced for a given *N*.

In conclusion, one can use the total site number *N* as a control parameter to design “useful” physical and thermodynamic (cycle) processes by keeping desirable observables constant and affecting others. It may provide valuable insights into the engineering of small (molecular) machines capable of performing specific tasks with high efficiency and precision. Further investigations of the discrete to continuous transition in state space and its potential impact on information-to-work conversion are avenues for future research. 

## Figures and Tables

**Figure 1 entropy-25-01218-f001:**
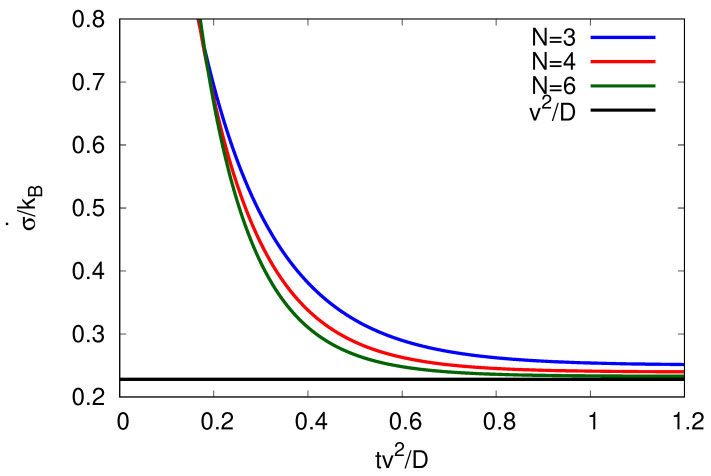
Entropy production rate $\dot{\sigma }\left(t\right)$ against time plotted for different *N*-site cycles by keeping the velocity *v* and diffusion coefficient *D* constant. The velocity *v* (here chosen to be $v=\frac{3D}{2\pi R}$) needs to be within the bounds implied by β≥0 of Equation (13) for N≥3. The steady-state value $v^{2}/D$ (black curve) is reached for N→∞.

**Figure 2 entropy-25-01218-f002:**
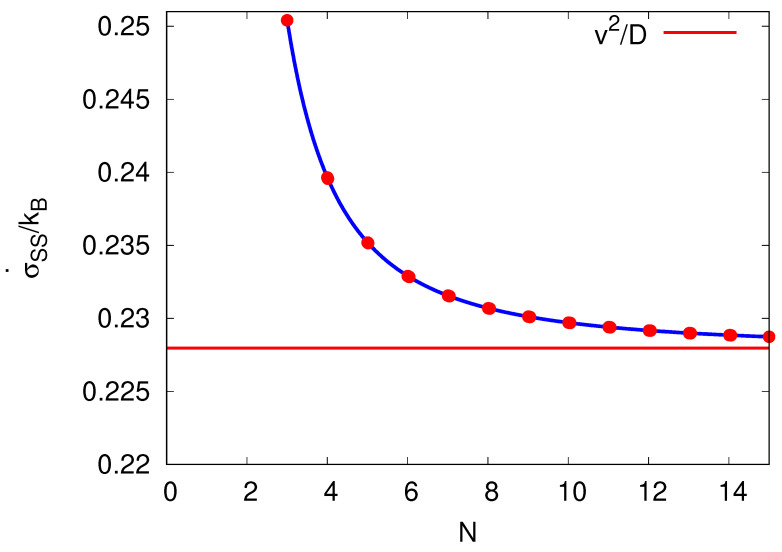
Entropy production rate at steady state $\dot{\sigma }\left(t\to \infty \right)\equiv {\dot{\sigma }}_{SS}\left[=-{\dot{S}}_{e}\right]$ against the number of sites *N* per cycle by keeping the velocity *v* and diffusion coefficient *D* constant. The velocity *v* (here chosen to be $v=\frac{3D}{2\pi R}$) needs to be within the bounds implied by β≥0 of Equation (13) for N≥3. The red line is the entropy production rate in the limit N→∞ and takes the value ${\dot{\sigma }}_{SS}\left(N\to \infty \right)=v^{2}/D$.

**Figure 3 entropy-25-01218-f003:**
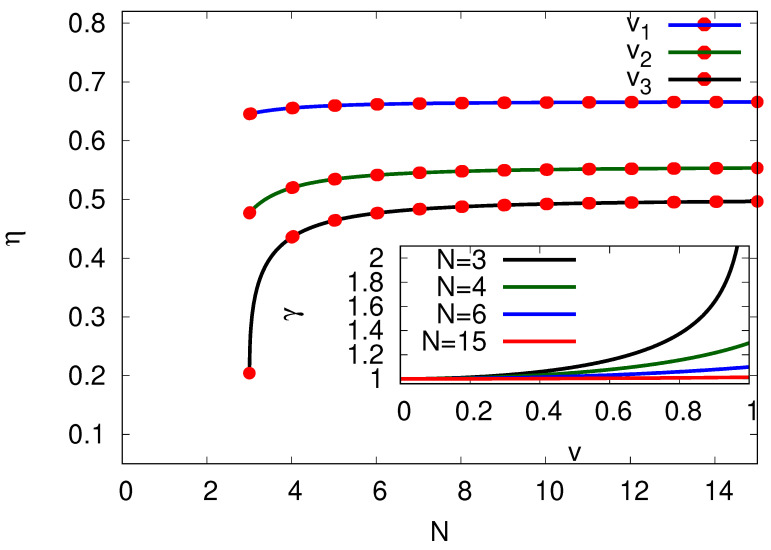
Efficiency at steady state η plotted against the number of sites *N* per cycle by keeping the velocity *v* and diffusion constant *D* constant. We depict η for $v_{1}=0.5v$, $v_{2}=0.8v$ and $v_{3}=v$, where we chose $v=\frac{3D}{\pi R}$ to be within the bounds implied by β≥0 of Equation (13) for N≥3. We chose the force $f\equiv vT{\dot{\sigma }}_{SS}\left(N\to \infty \right)=3k_{B}T/\pi R$. Inset: Damping constant $\gamma \equiv \gamma \left(v\right)D/\left(k_{B}T\right)=TD{\dot{\sigma }}_{SS}/\left(v^{2}k_{B}T\right)$ in dependence of the velocity for different *N*.

**Figure 4 entropy-25-01218-f004:**
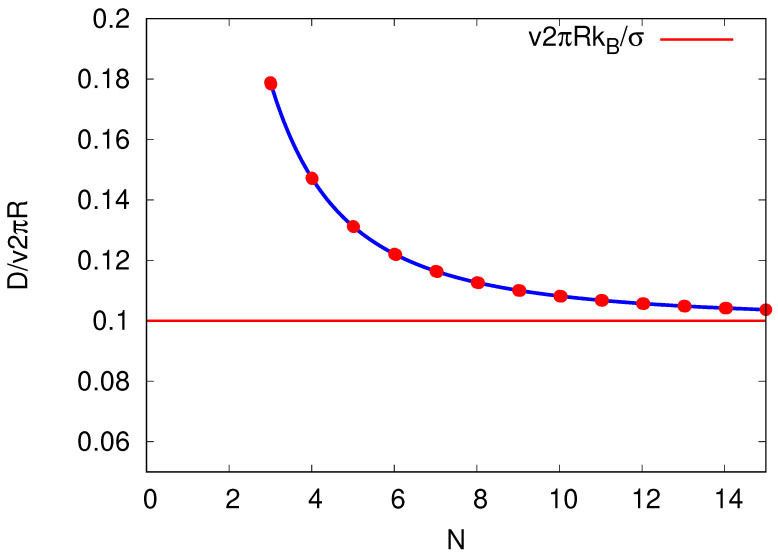
Diffusion constant *D* plotted against the number *N* of sites per cycle by keeping the entropy σ per cycle and velocity *v* constant for N≥3. We chose the produced entropy per cycle to be $\sigma /k_{B}=10$. The red line is the value $D\left(N\to \infty \right)\to v2\pi Rk_{B}/\sigma$ in the limit of N→∞.

**Figure 5 entropy-25-01218-f005:**
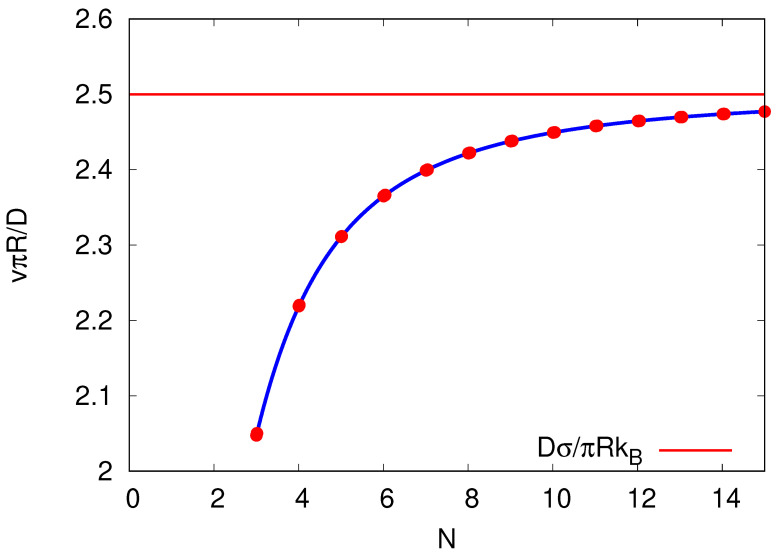
Velocity *v* plotted against the number *N* of sites per cycle by keeping the entropy σ per cycle and the diffusion coefficient *D* constant for N≥3. We chose the produced entropy per cycle to be $\sigma /k_{B}=5$. The red line is the value $v\left(N\to \infty \right)\to D\sigma /\pi Rk_{B}$ in the limit of N→∞.

**Figure 6 entropy-25-01218-f006:**
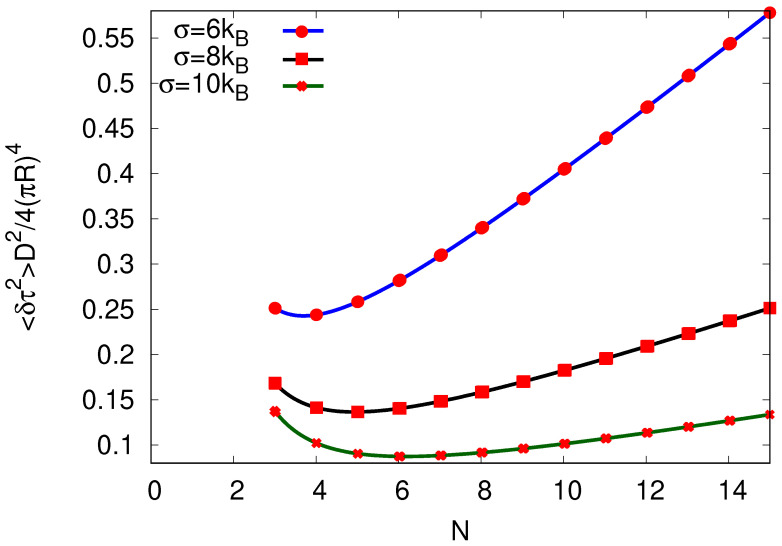
Variance in cycle completion time $\langle \delta \tau^{2}\rangle$ plotted against the number *N* of sites per cycle by keeping the entropy σ per cycle and the diffusion coefficient *D* constant for N≥3, while the velocity v(N) results from Equation (Equation 24). We show $\langle \delta \tau^{2}\rangle$ for three different choices of σ.

**Figure 7 entropy-25-01218-f007:**
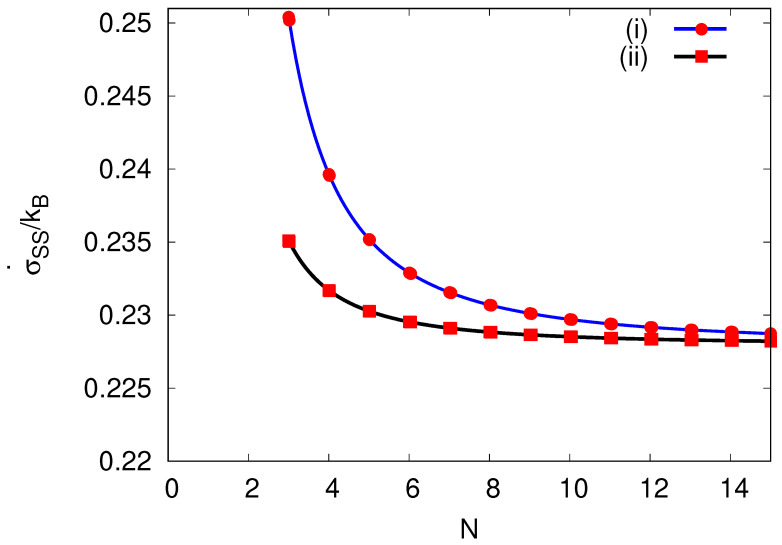
Entropy production rate at steady state, ${\dot{\sigma }}_{SS}$, against the number per cycle *N* by keeping the velocity *v* and diffusion coefficient *D* constant. Here, *v* (chosen to be $v=\frac{3D}{2\pi R}$) needs to be within the bound implied by β≥0 of Equation (38) for N≥3. (i) The blue circled line represents the entropy production rate in the continuous time limit Δt→0. (ii) The black squared line is the entropy production rate for a discrete time process using the maximal Δt allowed by Equation (Equation 39), $\Delta t=-\frac{D}{v^{2}}+\sqrt{\frac{D^{2}}{v^{2}}+\frac{\Delta x^{2}}{v^{2}}}$.

**Figure 8 entropy-25-01218-f008:**
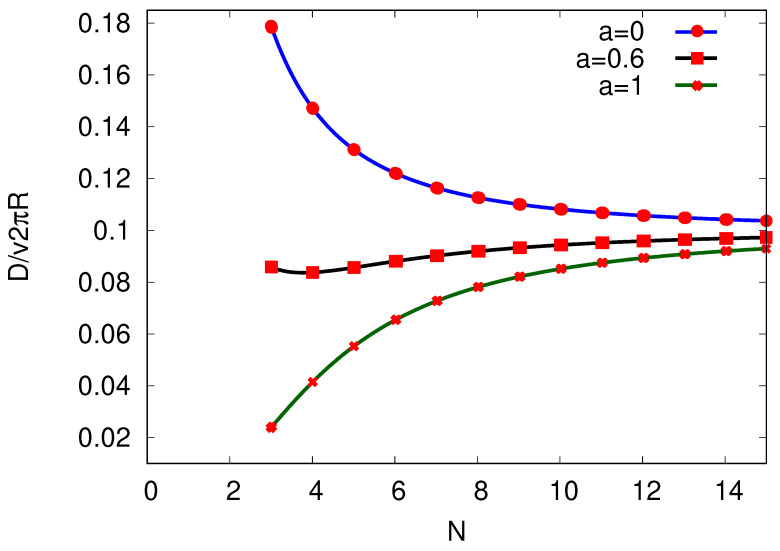
The diffusion coefficient *D* plotted against the number of sites *N* per cycle by keeping the entropy production per cycle σ and velocity *v* constant. The chosen time interval Δt for given *N* is restricted by the inequality (Equation 46). We take $\Delta t\left(a,N\right)=a\frac{\Delta x}{v}\tanh\left[\frac{\sigma }{2Nk_{B}}\right]$ (where 0≤a≤1) and depict D(a,N) with a=0 (blue curve), a=0.6 (black curve), and a=1 (green curve). We chose the produced entropy per cycle to be $\sigma /k_{B}=10$ for N≥3.

## Data Availability

The data presented in this study are available on request from the corresponding author.

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
