# Peer review of "Energy Conversion and Entropy Production in Biased Random Walk Processes—From Discrete Modeling to the Continuous Limit"

_entropy, 2023, doi:10.3390/e25081218_

Round 1
Reviewer 1 Report
In the manuscript 2544817, the authors study the energy conversion and entropy production. They compare a random walk discrete modeling to the continuous limit. They find that the motion becomes continuous in space and the time is not unique and depends on what physical observables are assumed to be unchanged in the process for the discrete model. However, one may choose to keep the speed and diffusion coefficient fixed during this limiting process. Moreover, the authors also study the processes in which the entropy production is kept constant at the cost of modified speed or diffusion coefficient.
Therefore, I recommend the manuscript to be published. However, the following issues should be fixed.
1. About the Eq. (2), it is actually a rate equation. The initial master equation for the system should be given, at least, some references should be given. Furthermore, the approximations should be clarified here.
2. Is it necessary for us to consider the positivity of the diffusion term D. In the stochastic view, it must be positive.
Author Response
See notes attached.

Reviewer 2 Report
The report is attached.

Author Response
See notes attached.

Round 2
Reviewer 2 Report
The authors have addressed my questions. I recommend its publication.